# Association between nocturia and serum albumin in the U.S. adults from NHANES 2005–2012

**Yangtao Jia**, **Fangzheng Yang**, **Xinke Dong**, **Libin Zhou***, **Huimin Long***

The Affiliated Lihuili Hospital of Ningbo University, Ningbo, Zhejiang, People's Republic of China

* lhllonghuimin@nbu.edu.cn (HL); zlburo2013@sina.com (LZ)

**Data Availability Statement:** The original dataset can be publicly accessed on the NHANES database (https://www.cdc.gov/nchs/nhanes/index.htm). The dataset and code used in our study are available for public viewing and download for free

## Abstract

### Background

Nocturia, a prevalent chronic condition, impacts individuals' quality of life but remains under-explored. This study aimed to assess the association between serum albumin levels and nocturia.

### Methods

Based on the analysis of the National Health and Nutrition Examination Survey (NHANES) database (2005–2012), our study included a total of 6345 adults ($\geq$20 years old). Nocturia was defined as $\geq$2 nocturnal voiding episodes. Logistic regression and smooth curve fitting analyzed the linear and nonlinear correlations between serum albumin and nocturia, with subgroup analysis.

### Results

Among 6345 participants, 1821 (28.7%) experienced nocturia. Logistic regression analysis revealed a linear negative correlation between serum albumin and nocturia risk (OR = 0.9549, 95% CI = 0.9280 ~ 0.9827, P = 0.002). Even after quartile division of serum albumin concentration, this correlation persisted within each group, and a smooth curve fitting validated the nonlinear negative correlation between the two. Subgroup analysis further demonstrated significant impacts of body mass index (BMI), alcohol consumption, and age on this association.

### Conclusion

This cross-sectional study indicated that higher serum albumin levels were associated with a reduced risk of nocturia in U.S. adults aged 20 and older, highlighting the importance of serum albumin in the prevention and treatment of nocturia and providing clinical guidance.

**Funding:** This work was supported by the Natural
Science Foundation of Ningbo Municipality
(2021J281), the Key Cultivating Discipline of
LihHuiLi Hospital (2022-P09) and Ningbo Key
Clinical Speciality Construction Project (2023-BZZ).

**Competing interests:** The authors have declared
that no competing interests exist.

# 1. Introduction

Nocturnal voiding, per the definition provided by the International Continence Society (ICS), pertains to the frequency of urination episodes experienced during the main sleep period [1]. Manifesting two or more times was considered clinically significant, as this criterion correlates with the quality of life and mortality [2, 3]. Increased nocturnal voiding was the most common clinical complain of lower urinary tract symptom (LUTS), accounting for 33% of all LUTS-related presentations. Its prevalence in young adults aged 20 to 40 years was 11.0%-35.2% in males and 20.4%-43.9% in females [4]. Furthermore, this prevalence doubled among older people [5, 6]. Additionally, nocturnal voiding was associated with various diseases, including sleep disorders, hypertension, diabetes, and cardiovascular diseases, as well as correlated with overall mortality [3, 4, 7].

Human Serum Albumin (HSA), typically present in concentrations ranging from 3.5 to 5 g/dl, constitutes 50%-60% of the total plasma proteins in healthy adults [8, 9]. As a vital carrier and drug-binding protein in plasma, HSA binds and transports both endogenous and exogenous molecules [10]. Additionally, albumin maintains balance among intracellular, extracellular and tissue fluids, maintaining osmotic pressure equilibrium, and plays a vital role in inflammation, antioxidant activity, endothelial stability and immune regulation [10–12].

Recent research indicated a higher prevalence of nocturnal voiding in liver cirrhosis patients exhibiting serum albumin levels below the general population average [13]. In patients with type 2 diabetes, those with overactive bladder (OAB) and nocturnal voiding had lower serum albumin levels compared to those without [14, 15]. These findings implied a certain degree of association between nocturnal voiding and serum albumin levels, although the exact mechanisms and direct relationship remained unclear. Moreover, cross-sectional studies with large samples investigating the current relationship between nocturnal voiding and albumin levels do not exist in populations. For these reasons, we conducted a comprehensive study to explore the relationship within a large population of adults aged 20 and above in the United States.

# 2. Methods

## 2.1 Population and design

The National Health and Nutrition Examination Survey (NHANES) was a biennial cross-sectional survey conducted in the United States by the National Center for Health Statistics. This survey systematically gathered representative data on US citizens' health and nutritional status to evaluate the population's overall health and nutritional well-being. Our research utilized integrated demographic, examination, laboratory, and questionnaire data from NHANES from 2005 to 2012. These comprehensive datasets were available for downloading on the NHANES website (https://www.cdc.gov/nchs/nhanes/index.htm). This study specifically explored the correlation between serum albumin levels and nocturia in NHANES participants from 2005 to 2012, involving a cohort of 40790 individuals. Participants were excluded if they were: (1) diagnosed with prostate cancer; (2) had prostate surgery; (3) had a rectal examination in the last seven days; (4) underwent prostate biopsy in the last four weeks; (5) diagnosed with infection or inflammation; (6) underwent cystoscopy in the last four weeks; (7) had missing data on nocturnal voiding or albumin; (8) had incomplete covariate information; (9) were under the age of 20. (10) pregnant and lactating women; (11) had used diuretics such as spironolactone, furosemide, and hydrochlorothiazide for one month. Following rigorous screening, 6345 participants were included in this study.

## 2.2 Exposure and outcome

Participants were screened for nocturia using a structured questionnaire. The inquiry posed was, "During the past 30 days, how many times per night did you most typically get up to urinate from the time going to bed at night to waking up in the morning?" The first urine passed on waking up was excluded from making the diagnosis of nocturia. Responses were stratified into categories: 0 times/night, 1 time/night, 2 times/night, 3 times/night, 4 times/night, and 5 or more times/night. In our investigation, individuals who reported urinating ≥2 times/night were identified as having nocturia.

Researchers conducted hypertension assessments using multiple approaches. Firstly, an affirmative response to the query, "Ever been told by a doctor or other health professional that you had hypertension, also called high blood pressure?" classified individuals as hypertensive. Secondly, participants who took prescriptions for hypertension were categorized as hypertensive. Lastly, individuals registering systolic blood pressure ≥130 mmHg or diastolic blood pressure ≥80 mmHg during the physical examination were also deemed hypertensive [16].

Participants were evaluated for diabetes through diverse methods. Firstly, an affirmative response to the question, "Ever been told by a doctor or health professional that you had diabetes or sugar diabetes?" led to classification as having diabetes. Secondly, participants reporting taking prescriptions for diabetes were labelled as having diabetes. Lastly, individuals with a fasting blood glucose level ≥12.6 mmol/L or a glycated hemoglobin level ≥6.5% in laboratory tests were also considered to have diabetes [17].

The Adult Treatment Panel III (ATP 3) of the National Cholesterol Education Program (NCEP) defines hyperlipidemia as having total cholesterol levels ≥200 mg/dL, triglycerides ≥150 mg/dL, High-density lipoprotein (HDL) levels ≤40 mg/dL in men or ≤50 mg/dL in women, or Low-density lipoprotein (LDL) levels ≥130 mg/dL [18, 19].

Additionally, depression was evaluated utilizing the nine-item Patient Health Questionnaire (PHQ-9) depression scale, where scores range from 0 to 27. A score of ≥10 indicates clinically significant depressive symptoms [20].

## 2.3 Covariates selection

We included potential confounding factors that may influence the association between nocturia and serum albumin levels in a multivariate model based on previous research and clinical experience [14, 21]. These factors encompass gender, age, race, education level, marital status, family poverty income ratio (PIR), body mass index (BMI), alcohol consumption, serum cotinine concentration, diabetes status, hypertension status, depressive disorder status, and hyperlipemia status. Participants' races were divided into five groups: Non-Hispanic White, Non-Hispanic Black, Mexican American, Other Hispanic, and Other races. Age (20–34 years, 35–49 years, 50–64 years, ≥65 years), education levels (Less than high school, High school graduate, More than high school), family PIR (<1, 1–4, >4), BMI (<25 kg/m2, 25–30 kg/m2, ≥30 kg/m2), and serum cotinine concentration (<0.015 ng/mL, 0.015–3 ng/mL, >3 ng/mL) were each classified into three categories. Gender (males, females), marital status (Married/Living with partner, Widowed/Divorced/Separated/Never married), diabetes history (Yes, No), hypertension history(Yes, No), alcohol consumption (Yes, No), hyperlipemia history (Yes, No), and depressive disorder history (Yes, No)were each categorized into two groups.

## 2.4 Statistical analysis

Because of the complex stratified and cluster sampling design of NHANES, we performed a weighted analysis. NHANES combined eight years of total survey data from 2005 to 2012. The

combined weight is expressed as $\text{WEIGHT}_{05\text{-}12} = (1/4) \times \text{WTMEC2YR}_{05\text{-}06} + (1/4) \times \text{WTMEC2YR}_{07\text{-}08} + (1/4) \times \text{WTMEC2YR}_{09\text{-}10} + (1/4) \times \text{WTMEC2YR}_{11\text{-}12}$

Baseline characteristics are reported with categorical variables and continuous variables. The former was presented with case numbers and percentages, and the latter was expressed as mean ± standard deviation. The baseline characteristics of subjects were categorized according to serum albumin quartiles. We used a multivariate logistic regression model to investigate the association between nocturia and diverse serum albumin levels. Association strength was quantified by odds ratios (OR) with corresponding 95% confidence intervals (CI). The multivariate model comprises model 1 (adjusted for none), model 2 (adjusted for age gender and race), and model 3 (adjusted for age, gender, race, marital status, education level, serum cotinine concentration, alcohol consumption, PIR status, BMI status, diabetes, hypertension, hyperlipemia and depressive disorder). Furthermore, we performed interaction and stratified analyses considering the listed covariates. Statistical analyses were conducted using STATA v16.0 and R project. Statistical significance was set at P<0.05.

### 2.5 Ethics statement

The research utilized data sourced from the publicly available National Health and Nutrition Examination Survey dataset (https://www.cdc.gov/nchs/nhanes/index.htm). Approval for the survey involving human subjects was granted by the National Center for Health Statistics Ethics Review Board. All participants were required to provide written informed consent.

## 3. Results

### 3.1. Selection of study population

Among the 40790 participants in NHANES from 2005 to 2012, 22692 individuals aged 20 and above participated in this study. Subsequently, 4206 samples with incomplete data on nocturia and albumin were excluded, leaving 18486 participants. After further excluding 3282 patients who met specific exclusion criteria, 15204 participants remained. Finally, samples lacking covariate information were removed, resulting in 6345 participants included in the study (Fig 1).

### 3.2 Baseline characteristics

Participants' characteristics were categorized into four groups according to quartiles of serum albumin concentration. Subsequently, the baseline characteristics of the participants were presented. Table 1 illustrated apparent variations in baseline characteristics among the albumin quartiles. The study included 6,345 adults, with a mean (SE) age of 46.355 years. The serum albumin concentration was 42.594 g/L. Of these, 3292 participants were male, and 3053 were female. Individuals with higher albumin concentrations were more likely to be male, aged 20–34 years, married or cohabiting, with a BMI < 25, and without diabetes, hypertension, or nocturia (Table 1).

### 3.3 Albumin was associated with a decrease in nocturia

Table 2 described the logistic regression analysis results between serum albumin concentration and nocturia. Initially, serum albumin was analyzed as a continuous variable, indicating a negative correlation with the risk of nocturia (OR = 0.9549, 95% CI = 0.9280–0.9827, P = 0.002). Subsequently, serum albumin concentration was divided into quartiles. Compared to quartile 1, individuals in the other quartiles had a decreased odds of experiencing nocturia by 33.07% (OR = 0.7693, 95% CI = 0.6220–0.9514, P = 0.016), 28.98% (OR = 0.7102, 95% CI = 0.5716–

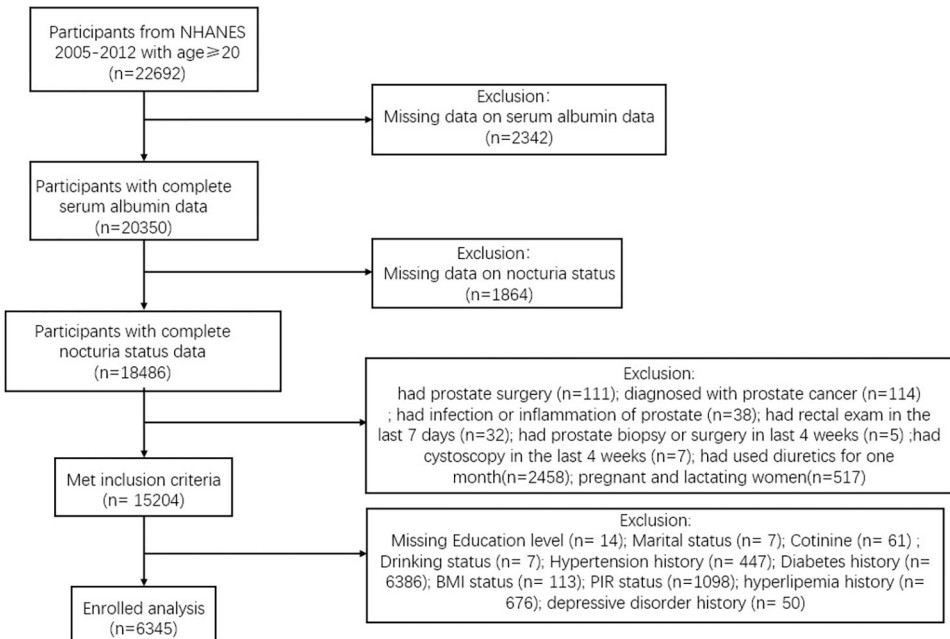

**Fig 1. Flow chart illustrating the selection of participants in NHANES from 2005 to 2012.**

0.8825, P = 0.002), and 25.04% (OR = 0.7496, 95% CI = 0.5900–0.9523, P = 0.018), respectively. In all models, serum albumin exhibited a negative correlation with nocturia, with a significant decrease in the risk of nocturia as serum albumin levels increased. Additionally, restricted cubic spline regression was used to assess the dose-response relationship between serum albumin and nocturia in the three models (Fig 2). The study results showed a significant nonlinear relationship between albumin and nocturia, indicating that higher serum albumin levels were associated with a lower risk of nocturia (P < 0.0001, P for non-linear < 0.0001).

## 3.4 Subgroup analysis

Subgroup analyses were conducted to assess the strength and consistency of the association between serum albumin concentration and nocturia (Fig 3). In model 3 accounting for all relevant adjustments, the correlation between nocturia and serum albumin exhibited significant differences in age, drink and BMI interaction tests (P<0.05). Similarly, subgroup analysis of the correlation between serum albumin concentration and nocturia frequency was conducted in model 3 (Fig 4), revealing significant differences across different levels of PIR subgroups (P = 0.01).

## 4. Discussion

In this study, we enrolled 22692 participants aged 20 years and older to investigate the correlation between serum albumin and nocturia. Although previous studies had explored the aetiology of nocturia, there had been limited research on the association between nocturia and serum albumin. Our results revealed a negative correlation between serum albumin levels and nocturia (urination ≥ 2 times/night), whether treated as a continuous or quartile variable. Importantly, this association remained altered by age BMI and drink. This was the inaugural cross-sectional study employing the NHANES database to scrutinize the relationship between nocturia and serum albumin in a large and diverse population.

**Table 1. Characteristics of participants with different serum albumin levels in NHANES from 2005 to 2012.**

| | Q1 | Q2 | Q3 | Q4 | total | P-vaule |
|---|---|---|---|---|---|---|
| **Age(years)** | | | | | | 0.928 |
| | 48.533±0.439 | 48.572±0.485 | 47.113±0.459 | 41.833±0.443 | 46.355±0.231 | |
| **Serum albumin(g/L)** | | | | | | <0.001 |
| | 38.545±0.050 | 41.557±0.017 | 43.451±0.016 | 46.341±0.047 | 42.594±0.049 | |
| **Gender(n%)** | | | | | | <0.001 |
| Male | 581 (33.18%) | 656 (44.93%) | 919 (59.33%) | 1136 (71.67%) | 3292 (51.88%) | |
| Female | 1170 (66.82%) | 804 (55.07%) | 630 (40.67%) | 449 (28.33%) | 3053 (48.12%) | |
| **Age(n%)** | | | | | | <0.001 |
| 20–34 | 336 (19.19%) | 303 (20.75%) | 365 (23.56%) | 608 (38.36%) | 1612 (25.41%) | |
| 35–49 | 513 (29.3%) | 396 (27.12%) | 444 (28.66%) | 408 (25.74%) | 1761 (27.75%) | |
| 50–64 | 504 (28.78%) | 416 (28.49%) | 443 (28.6%) | 373 (23.53%) | 1736 (27.36%) | |
| ≥65 | 398 (22.73%) | 345 (23.63%) | 297 (19.17%) | 196 (12.37%) | 1236 (19.48%) | |
| **Race/Ethnicity(n%)** | | | | | | <0.001 |
| Mexican American | 283 (16.16%) | 264 (18.08%) | 271 (17.5%) | 306 (19.31%) | 1124 (17.71%) | |
| Other Hispanic | 176 (10.05%) | 131 (8.97%) | 157 (10.14%) | 148 (9.34%) | 612 (9.65%) | |
| Non-Hispanic White | 768 (43.86%) | 695 (47.6%) | 774 (49.97%) | 763 (48.14%) | 3000 (47.28%) | |
| Non-Hispanic Black | 444 (25.36%) | 279 (19.11%) | 231 (14.91%) | 196 (12.37%) | 1150 (18.12%) | |
| Other Race—Including Multi-Racial | 80 (4.57%) | 91 (6.23%) | 116 (7.49%) | 172 (10.85%) | 459 (7.23%) | |
| **Education(n%)** | | | | | | 0.017 |
| Less than high school | 527 (30.1%) | 393 (26.92%) | 385 (24.85%) | 407 (25.68%) | 1712 (26.98%) | |
| High school | 401 (22.9%) | 333 (22.81%) | 367 (23.69%) | 355 (22.4%) | 1456 (22.95%) | |
| More than high school | 401 (22.9%) | 333 (22.81%) | 367 (23.69%) | 355 (22.4%) | 1456 (22.95%) | |
| **Marital status(n%)** | | | | | | <0.001 |
| Married/Living with partner | 807 (46.09%) | 564 (38.63%) | 561 (36.22%) | 604 (38.11%) | 2536 (39.97%) | |
| Widowed/Divorced/Separated/Never married | 944 (53.91%) | 896 (61.37%) | 988 (63.78%) | 981 (61.89%) | 3809 (60.03%) | |
| **PIR(n%)** | | | | | | <0.001 |
| ≤1 | 451 (25.76%) | 304 (20.82%) | 288 (18.59%) | 343 (21.64%) | 1386 (21.84%) | |
| 1–4 | 913 (52.14%) | 784 (53.7%) | 815 (52.61%) | 828 (52.24%) | 3340 (52.64%) | |
| ≥4 | 387 (22.1%) | 372 (25.48%) | 446 (28.79%) | 414 (26.12%) | 1619 (25.52%) | |
| **BMI(n%)** | | | | | | <0.001 |
| <25 | 301 (17.19%) | 371 (25.41%) | 476 (30.73%) | 664 (41.89%) | 1812 (28.56%) | |
| 25–30 | 493 (28.16%) | 515 (35.27%) | 586 (37.83%) | 580 (36.59%) | 2174 (34.26%) | |
| ≥30 | 957 (54.65%) | 574 (39.32%) | 487 (31.44%) | 341 (21.51%) | 2359 (37.18%) | |
| **Cotinine(ng/mL) (n%)** | | | | | | 0.041 |
| <0.015 | 338 (19.3%) | 323 (22.12%) | 349 (22.53%) | 337 (21.26%) | 1347 (21.23%) | |
| 0.015–3 | 909 (51.91%) | 730 (50%) | 789 (50.94%) | 759 (47.89%) | 3187 (50.23%) | |
| >3 | 504 (28.78%) | 407 (27.88%) | 411 (26.53%) | 489 (30.85%) | 1811 (28.54%) | |
| **Drink at least 12 drinks/year(n%)** | | | | | | <0.001 |
| No | 595 (33.98%) | 402 (27.53%) | 353 (22.79%) | 292 (18.42%) | 1642 (25.88%) | |
| Yes | 1156 (66.02%) | 1058 (72.47%) | 1196 (77.21%) | 1293 (81.58%) | 4703 (74.12%) | |
| **Diabetes history(n%)** | | | | | | <0.001 |
| No | 1228 (70.13%) | 1108 (75.89%) | 1256 (81.08%) | 1348 (85.05%) | 4940 (77.86%) | |
| Yes | 523 (29.87%) | 352 (24.11%) | 293 (18.92%) | 237 (14.95%) | 1405 (22.14%) | |
| **Hypertension history(n%)** | | | | | | <0.001 |
| No | 712 (40.66%) | 661 (45.27%) | 707 (45.64%) | 828 (52.24%) | 2908 (45.83%) | |
| Yes | 1039 (59.34%) | 799 (54.73%) | 842 (54.36%) | 757 (47.76%) | 3437 (54.17%) | |
| **Hyperlipemia(n%)** | | | | | | <0.001 |

*(Continued)*

**Table 1.** (Continued)

| | Q1 | Q2 | Q3 | Q4 | total | P-vaule |
|---|---|---|---|---|---|---|
| **No** | 372 (21.25%) | 401 (27.47%) | 404 (26.08%) | 465 (29.34%) | 1642 (25.88%) | |
| **Yes** | 1379 (78.75%) | 1059 (72.53%) | 1145 (73.92%) | 1120 (70.66%) | 4703 (74.1%) | |
| **Depressive disorder(n%)** | | | | | | <0.001 |
| **No** | 1545 (88.24%) | 1324 (90.68%) | 1439 (92.9%) | 1489 (93.94%) | 5797 (91.36%) | |
| **Yes** | 206 (11.76%) | 136 (9.32%) | 110 (7.1%) | 96 (6.06%) | 548 (8.64%) | |
| **Nocturia(n%)** | | | | | | <0.001 |
| **Yes** | 1078 (61.56%) | 1037 (71.03%) | 1154 (74.5%) | 1255 (79.18%) | 4524 (71.3%) | |
| **No** | 673 (38.44%) | 423 (28.97%) | 395 (25.5%) | 330 (20.82%) | 1821 (28.7%) | |
| **Times of nocturia(n%)** | | | | | | <0.001 |
| **0** | 459 (26.21%) | 450 (30.82%) | 529 (34.15%) | 680 (42.9%) | 2118 (33.38%) | |
| **1** | 619 (35.35%) | 587 (40.21%) | 625 (40.35%) | 575 (36.28%) | 2406 (37.92%) | |
| **2** | 376 (21.47%) | 261 (17.88%) | 254 (16.4%) | 213 (13.44%) | 1104 (17.4%) | |
| **3** | 172 (9.82%) | 113 (7.74%) | 93 (6%) | 74 (4.67%) | 452 (7.12%); | |
| **4** | 77 (4.4%) | 30 (2.05%) | 31 (2%) | 24 (1.51%) | 162 (2.55%); | |
| **5 or more** | 48 (2.74%) | 19 (1.3%) | 17 (1.1%) | 19 (1.2%) | 103 (1.62%) | |

Q1: Serum albumin<40g/L, Q2:40g/L≤PIR<42g/L, Q3: 42g/L≤PIR<44g/L, Q4: PIR>44g/L.

There was a growing awareness of nocturia as a highly prevalent chronic condition, meriting in-depth investigation. However, existing treatment modalities were constrained, and diagnostic markers were scarce, resulting in suboptimal treatment and management outcomes for nocturia [1, 22]. With the progress in nocturia research, several studies had found a significant association between serum albumin and nocturia [13]. Based on these insights, we conducted a study examining the correlation between albumin and nocturia utilizing the NHANES database. Our findings aligned with prior reports, affirming a negative correlation between serum albumin levels and nocturia [14]. In a recent clinical association involving patients with chronic liver cirrhosis in Taiwan, participants exhibited an average serum albumin level of 3.85 g/dl, below the norm for the general population, with 62% of these participants displaying noticeable symptoms of nocturia [13]. In a preceding cross-sectional association on type II diabetes, participants experiencing nocturia exhibited markedly lower serum albumin levels than their counterparts without nocturia [14]. Notably, in additional clinical exploration of type II diabetes, individuals with overactive bladder (OAB) manifested a substantial reduction in serum albumin levels relative to those without OAB [15]. Drawing

**Table 2. Association between nocturia and serum albumin.**

| | Model 1 | P value | Model 2 | P value | Model 3 | P value |
|---|---|---|---|---|---|---|
| **Serum albumin** | 0.8949(0.8739–0.9165) | <0.0001 | 0.9294 (0.9040–0.9555) | <0.0001 | 0.9549 (0.9280–0.9827) | 0.002 |
| **Serum albumin** | | | | | | |
| **Q1** | Reference | | Reference | | Reference | |
| **Q2** | 0.6584(0.5423–0.7992) | <0.0001 | 0.6811 (0.5542–0.8371) | 0.0003 | 0.7693 (0.6220–0.9514) | 0.016 |
| **Q3** | 0.5181(0.4265–0.6295) | <0.0001 | 0.5925 (0.4805–0.7305) | <0.0001 | 0.7102 (0.5716–0.8825) | 0.002 |
| **Q4** | 0.4373(0.3549–0.5384) | <0.0001 | 0.6120 (0.4855–0.7715) | <0.0001 | 0.7496 (0.5900–0.9523) | 0.018 |

Model 1 adjusts for none.

Model 2 adjusts for age, ethnicity, and gender.

Model 3 adjusts for age, gender, ethnicity, education, BMI, marital status, PIR status, serum cotinine, alcohol status, hypertension history, diabetes history, hyperlipemia history and depressive disorder history.

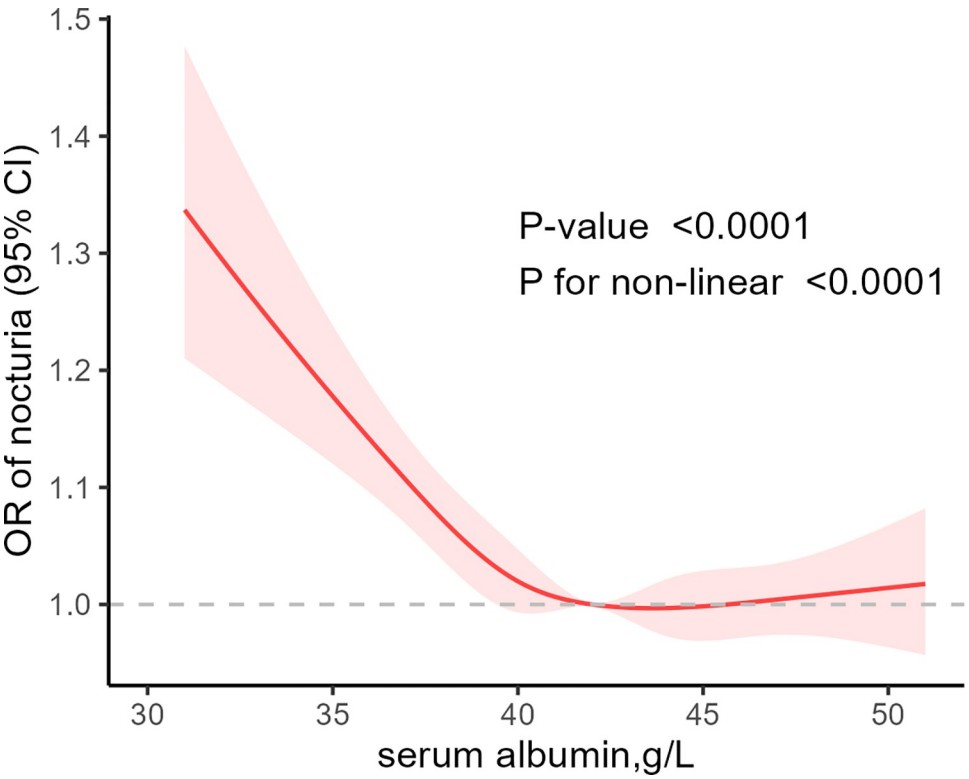

**Fig 2. Relationship between nocturia and albumin in model 3.**

from these findings, we posited that serum albumin could indicate the overall state of nocturia, presenting potential as a diagnostic marker in clinical settings and offering guidance for treatment.

However, the precise mechanism linking serum albumin and nocturia remained unclear. We investigated potential pathways by reviewing existing literature. It was widely acknowledged that as the most abundant protein in plasma, albumin played a pivotal role in diverse physiological functions, such as pH regulation in plasma and the modulation of fluid transport [22, 23]. Prior investigations had partially revealed the mechanisms associating albumin levels with nocturia. Diminished serum albumin levels could reduce osmotic pressure and fluid accumulation in various bodily regions. Thus, the human body initiates compensatory diuresis to uphold the equilibrium of colloid osmotic pressure [24]. Also, maintaining a supine position at night allowed the re-entry of accumulated fluid into circulation, causing an elevation in systolic blood pressure. Concurrently, there was a reduction in serum levels of nocturnal antidiuretic hormones, resulting in increased urine output [25]. Secondly, serum albumin levels clinically served as a crucial indicator for evaluating liver function [26, 27]. Diminished serum albumin levels signified a reduction in hepatic protein synthesis capacity. Historical research indicated a close correlation between cholinesterase and albumin levels produced by the liver [27, 28]. The decline in serum albumin levels signified a decrease in cholinesterase content and implied an increase in cholinergic activity. Additionally, activating the micturition reflex delayed the clearance of a substantial amount of acetylcholine secreted by parasympathetic nerve terminals in the bladder, promoting bladder contraction and voiding [29]. These mechanisms proposed that serum albumin might impact bladder spasms and voiding frequency, consequently influencing nocturnal urination. Based on these potential mechanisms and our

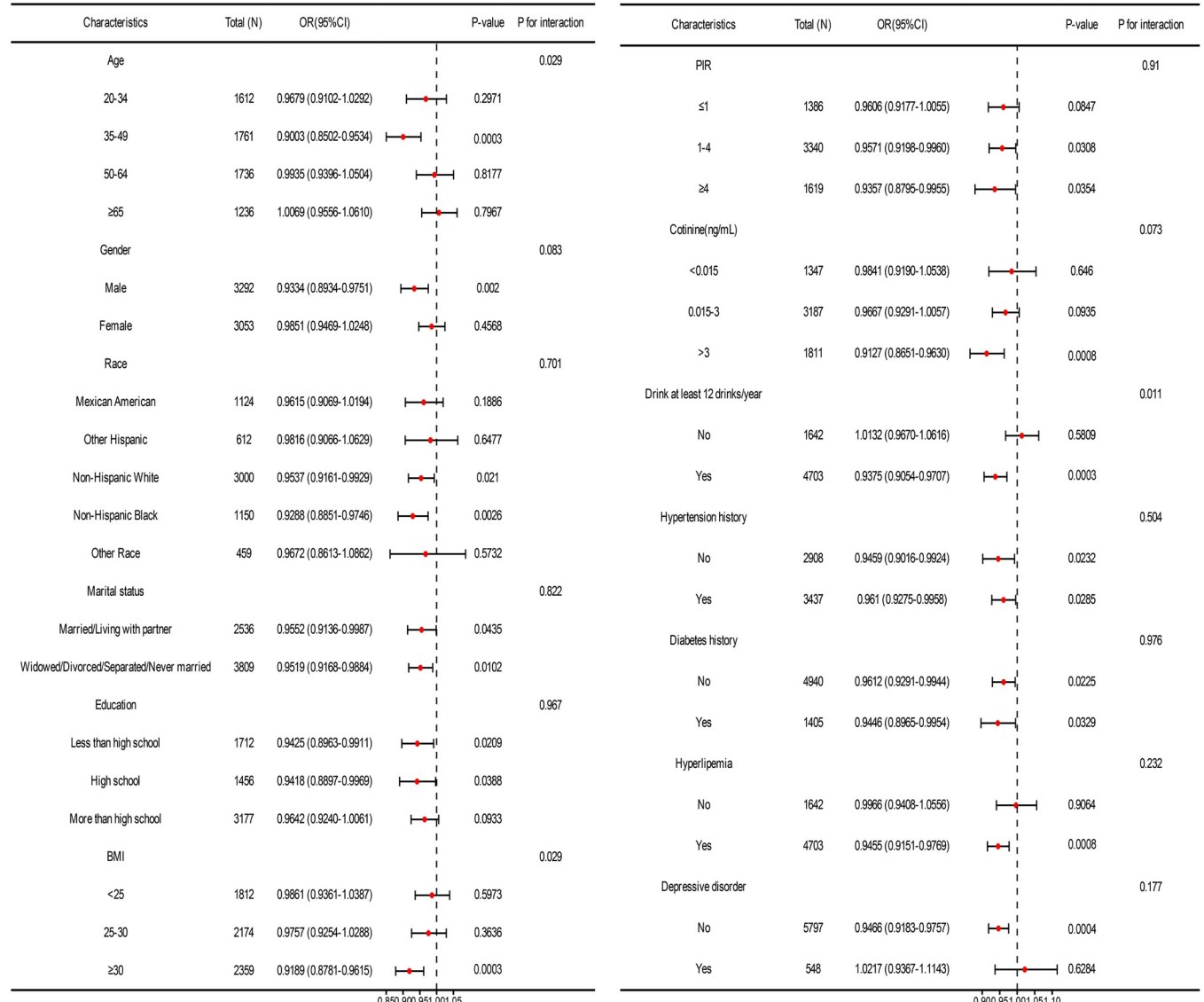

**Fig 3. Associations of nocturia and serum albumin in various subgroups.**

study findings, increasing patients' serum albumin levels may be beneficial in alleviating nocturia symptoms in clinical practice.

Through subgroup analysis, we found significant interactions between BMI, alcohol consumption, and age in the correlation between nocturia and serum albumin. Previous research and clinical expertise also indicated that age was a significant factor in nocturia, as the prevalence of diseases such as diabetes, obesity, and hypertension increased with age, affecting urinary function and consequently increasing the risk of nocturia [30, 31]. These findings validated the credibility of our results. Furthermore, individuals who consumed alcohol exhibited a lower risk of nocturia, possibly due to the improvement of insulin sensitivity and reduction in testosterone levels with mild to moderate alcohol consumption, consistent with previous research [32, 33]. However, other studies suggested that moderate to heavy alcohol consumption led to more severe lower urinary tract symptoms. contrary to our experimental conclusions [34, 35]. Additionally, previous research indicated that obesity was also a

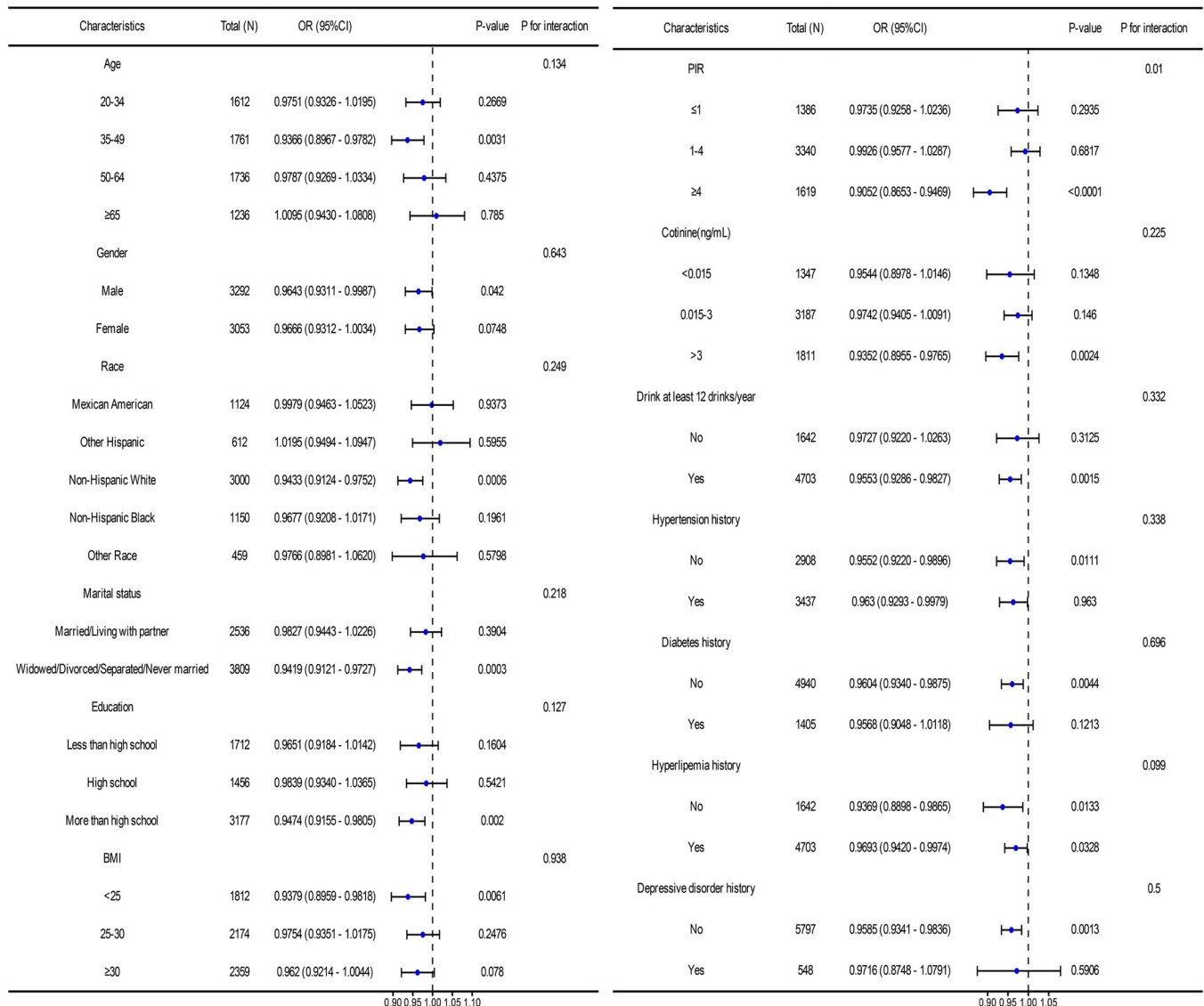

**Fig 4. Association of nocturia frequency and serum albumin in different subpopulations.**

significant influencing factor in nocturia. Obesity could lead to increased intra-abdominal pressure, directly causing nocturia [36]. Insulin resistance and hyperinsulinemia caused by obesity could stimulate hypothalamic centers, regulating sympathetic tone, increasing cate-cholamine levels, promoting prostate growth, and indirectly causing nocturia [37, 38]. Studies in Taiwan and Japan had shown a positive correlation between obesity and the risk of nocturia [39, 40]. Moreover, we further analyzed the frequency of nocturia and found significant inter-actions between PIR and the correlation between nocturia frequency and serum albumin. PIR was an essential indicator of income level, and lower household income was associated with poorer household health protection, hindering timely prevention, diagnosis, and treatment of diseases [41]. Many studies also suggested a significant correlation between income level and nocturia [42]. However, due to the cross-sectional nature of the NHANES database, the exact mechanisms by which these factors influenced the association between serum albumin and nocturia still needed to be determined.

This study had several significant strengths. Firstly, using a large sample size enhanced the generalizability of our research findings. Secondly, we conducted a preliminary investigation into the correlation between nocturia and serum albumin using the NHANES database, thereby improving physicians' understanding of serum albumin levels in patients with nocturia. Lastly, we systematically stratified various factors and examined their interactions with nocturia. However, this study had some limitations. Firstly, the assessment of participants' nocturia relied on self-reported questionnaires, and diagnosis depended on participants completing voiding diaries, which were subjective and may have needed to be more accurate. Secondly, our sample size was relatively small, and larger-scale studies were needed in the future to obtain more precise results. Additionally, despite the inclusion of many confounding factors and exclusion criteria, other unconsidered factors may still have affected the experimental results. Furthermore, this study was cross-sectional, only suitable for exploring associations, and could not infer causality. Moreover, based on the current research findings, the exact mechanism by which serum albumin affected nocturia remained unclear, and more prospective studies were needed to validate our hypotheses and discuss the mechanism between nocturia and serum albumin.

## 5. Conclusion

The association between serum albumin levels and the risk of nocturia is identified. A decrease in serum albumin levels correlates with an increased risk of nocturia, suggesting it can be a good indicator for predicting the risk. Although we provide some clinical insights, additional randomized controlled studies will be necessary in the future to provide more evidence for precise prevention and treatment.

## Acknowledgments

We extend our gratitude to all participants in the US National Health and Nutrition Examination Survey (NHANES) and value the unrestricted access offered to the public by the NHANES database.

## Author Contributions

**Data curation:** Yangtao Jia, Fangzheng Yang, Libin Zhou.

**Funding acquisition:** Huimin Long.

**Software:** Yangtao Jia, Fangzheng Yang, Xinke Dong.

**Visualization:** Xinke Dong.

**Writing – original draft:** Yangtao Jia, Libin Zhou.

**Writing – review & editing:** Yangtao Jia, Libin Zhou, Huimin Long.

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
