## [Decision Letter · Decision Letter 0]

26 Jul 2024

PONE-D-24-23292Association between nocturia and serum albumin in the U.S. adults from NHANES 2005-2012PLOS ONE

Dear Dr. Jia,

Thank you for submitting your manuscript to PLOS ONE. After careful consideration, we feel that it has merit but does not fully meet PLOS ONE’s publication criteria as it currently stands. Therefore, we invite you to submit a revised version of the manuscript that addresses the points raised during the review process.

**Thanks for sending the article to the PLOS ONE One journal.**

**Line 117–120 State that participants were screened for nocturia using a structured questionnaire. The inquiry posed was, "During the past 30 days, how many times per night did you most typically get up to urinate from the time going to bed at night to waking up in the morning?"**

**It is not clear whether the first urine passed on waking up was included or excluded from making the diagnosis of nocturia. There is a need to clarify this.**

We look forward to receiving your revised manuscript.

Kind regards,

Innocent Ijezie Chukwuonye, MBBS, FMCP(Internal Medicine)

Academic Editor

PLOS ONE

Journal Requirements:

1. When submitting your revision, we need you to address these additional requirements. Please ensure that your manuscript meets PLOS ONE's style requirements, including those for file naming. The PLOS ONE style templates can be found at https://journals.plos.org/plosone/s/file?id=wjVg/PLOSOne_formatting_sample_main_body.pdf and https://journals.plos.org/plosone/s/file?id=ba62/PLOSOne_formatting_sample_title_authors_affiliations.pdf 2. Thank you for stating in your Funding Statement: This work was supported by the
Natural
Science Foundation of
Ningbo Municipality (2021J281), the Key Cultivating Discipline of LihHuiLi Hospital (2022-P09) and Ningbo Key Clinical Speciality Construction Project (2023-BZZ)  Please provide an amended statement that declares all the funding or sources of support (whether external or internal to your organization) received during this study, as detailed online in our guide for authors at http://journals.plos.org/plosone/s/submit-now.  Please also include the statement “There was no additional external funding received for this study.” in your updated Funding Statement. Please include your amended Funding Statement within your cover letter. We will change the online submission form on your behalf. 3. Thank you for uploading your study's underlying data set. Unfortunately, the repository you have noted in your Data Availability statement does not qualify as an acceptable data repository according to PLOS's standards. At this time, please upload the minimal data set necessary to replicate your study's findings to a stable, public repository (such as figshare or Dryad) and provide us with the relevant URLs, DOIs, or accession numbers that may be used to access these data. For a list of recommended repositories and additional information on PLOS standards for data deposition, please see https://journals.plos.org/plosone/s/recommended-repositories. 4. When completing the data availability statement of the submission form, you indicated that you will make your data available on acceptance. We strongly recommend all authors decide on a data sharing plan before acceptance, as the process can be lengthy and hold up publication timelines. Please note that, though access restrictions are acceptable now, your entire data will need to be made freely accessible if your manuscript is accepted for publication. This policy applies to all data except where public deposition would breach compliance with the protocol approved by your research ethics board. If you are unable to adhere to our open data policy, please kindly revise your statement to explain your reasoning and we will seek the editor's input on an exemption. Please be assured that, once you have provided your new statement, the assessment of your exemption will not hold up the peer review process. 5. Please include your full ethics statement in the ‘Methods’ section of your manuscript file. In your statement, please include the full name of the IRB or ethics committee who approved or waived your study, as well as whether or not you obtained informed written or verbal consent. If consent was waived for your study, please include this information in your statement as well.  6. Please include captions for your Supporting Information files at the end of your manuscript, and update any in-text citations to match accordingly. Please see our Supporting Information guidelines for more information: http://journals.plos.org/plosone/s/supporting-information.  

Reviewers' comments:

Reviewer's Responses to Questions

**Comments to the Author**

1. Is the manuscript technically sound, and do the data support the conclusions?

Reviewer #1: Yes

Reviewer #2: Yes

Reviewer #3: Yes

2. Has the statistical analysis been performed appropriately and rigorously? 

Reviewer #1: Yes

Reviewer #2: I Don't Know

Reviewer #3: Yes

3. Have the authors made all data underlying the findings in their manuscript fully available?

Reviewer #1: Yes

Reviewer #2: Yes

Reviewer #3: Yes

4. Is the manuscript presented in an intelligible fashion and written in standard English?

Reviewer #1: Yes

Reviewer #2: Yes

Reviewer #3: Yes

5. Review Comments to the Author

**Reviewer #1: **Figure 2 Relationship between nocturia and albumin in Model 3. Mention it paragraph.

Abbreviation in abstract without clarify. Rewrite them with journal style.

References rewrite them with journal style too.

**Reviewer #2:** The subject has been previously investigated but the sample size and cosidering many factors that might affect nocturia makes the approach a novel one. The clinical impact of the study is not clear for me and its influence on patients management is not fully understood but it opens the door for further studies.

Line 66 refernce 2: the refernce shows only correlation with th QOL and not motality.

Line 67 I suggest using complain or presentation rather than symptom which was repeated in the next line.

Line 86 reference 14; The patients with OAB have significantly lower SHIM score, testosterone level, and serum albumin level, have more proportion of severe ED. Lower not higher.

Line 201 do you mean exclusion criteria?

Line 370 I think good is more appropriate than excellent

**Reviewer #3: **The manuscript being presented in its current format is acceptable for publication in the journal PLOS ONE.

The findings of the study and the thorough explanation of this particular topic is adequate.

6. PLOS authors have the option to publish the peer review history of their article (what does this mean?). If published, this will include your full peer review and any attached files.

Reviewer #1: **Yes: **Abudea Y. A. A.

Reviewer #2: **Yes: **Aly A. Abdel-Rahim

Reviewer #3: No

---

## [Author Response · Author response to Decision Letter 0]

1 Aug 2024

We would like to sincerely thank the editor and reviewers for their thorough evaluation of my manuscript. Their valuable feedback and constructive criticism have played a crucial role in refining the quality of the article. I am grateful for their dedicated efforts during the review process and for taking their comments seriously and making responsible revisions to my work.

Editorial comment

Question 1：Line 117–120 State that participants were screened for nocturia using a structured questionnaire. The inquiry posed was, "During the past 30 days, how many times per night did you most typically get up to urinate from the time going to bed at night to waking up in the morning? “It is not clear whether the first urine passed on waking up was included or excluded from making the diagnosis of nocturia. There is a need to clarify this.

Our response：We have included additional details to clarify that "the first urine passed on waking up was excluded from making the diagnosis of nocturia," which enhances the clarity of our diagnostic criteria and prevents ambiguity. (line 102-103)

Question 2: Line21: Delete keyword “cross-section study”

Our response: we removed the keyword to make the article's key points more prominent. (line 20)

Question 3: Line40, line 281: Replace the word "link" with "association."

Our response: We have revised the title to better align with the journal's formatting requirements. (line 24, line 264)

Question 4：line94: change the title "Materials and Methods" to "Methods."

Our response：We have revised the title to better align with the journal's formatting requirements. (line 77)

Question 5: line 145-146, line 331: Write the meaning of HDL, LDL and LUTS before the abbreviation.

Our response: We have corrected these errors and the explanation for the abbreviation "LUTS" has been provided in the introduction section. (line 52, line 125-126). Additionally, we made similar revisions in the abstract to better align with academic writing standards. (line 27, line 38)

Question 6: line 156, line 268, line 280, line 283, Reference the previous research

Our response: Based on previous research results, we found a significant association between serum albumin and nocturia. (DOI: 10.3390/jcm10132838, 10.1111/j.1743-6109.2012.02738.x) (line 136) Additionally, we incorporated confounding factors into our study by referencing previous research on nocturia in the NHANES database. (DOI: 10.3389/fcvm.2023.1165092, 10.3389/fpubh.2023.1186848) Furthermore, we cited relevant literature to provide a stronger foundation for our discussion. (DOI: 10.1111/j.1743-6109.2012.02738.x, 10.3390/jcm10132838) (line 264, line 267)

Question 7: Line 284, line287: Replace the word " investigation" with "association."

Our response: We have revised the vocabulary to make our expression more precise and better aligned with academic standards. (line 267, line 271)

Question 8: Line353: Replace the word "several" with "some."

Our response: We have revised the vocabulary as requested to ensure a more objective and clear expression. (line 333)

Review Comments 

Reviewer 1: 

Question 1：Figure 2 Relationship between nocturia and albumin in Model 3. Mention it paragraph.

Our response: We have specifically described the negative correlation between nocturia and serum albumin in Figure 2 within the relevant paragraphs, making our results more visually intuitive (lines 219-222).

Question 2: Abbreviation in abstract without clarify. Rewrite them with journal style. References rewrite them with journal style too.

Our response: We have clarified the abbreviations in the abstract and updated the format of the References section to meet the requirements of PLOS ONE. (lines 27, 38, line 373-499).

Reviewer 2

Question 1: Line 66 reference 2: the reference shows only correlation with the QOL and not mortality.

Our response: We apologize for the oversight in not including relevant references in our manuscript. We have now cited the appropriate literature to support our expressions (DOI: 10.1016/j.juro.2010.09.108). (line 51)

Question 2: Line 67 I suggest using complain or presentation rather than symptom which was repeated in the next line.

Our response: We have revised "symptom" to "complain" as requested, which improves the clarity and coherence of the expression. (line 52)

Question 3：Line 86 reference 14; The patients with OAB have significantly lower SHIM score, testosterone level, and serum albumin level, have more proportion of severe ED. Lower not higher.

Our response: We apologize for this oversight. We have corrected the error by changing "higher" to "lower" to accurately reflect the results of previous study. (line 68)

Question 4：Line 201 do you mean exclusion criteria?

Our response: We apologized for the lack of clarity in our expressions. The term "criteria" refers to the exclusion criteria. We have revised the text to avoid ambiguity (lines 183-184). The specific exclusion criteria are detailed in lines 89-95.

Question 5：Line 370 I think good is more appropriate than excellent.

Our response: We have revised the vocabulary, which has made our expression more objective. (line 349)

Reviewer 3

Question 1: The manuscript being presented in its current format is acceptable for publication in the journal PLOS ONE. The findings of the study and the thorough explanation of this particular topic is adequate.

Our response: Thanks for your positive feedback. We are pleased to hear that the manuscript is acceptable for publication in PLOS ONE and that the findings and explanation of the topic meet the journal's standards.

---

## [Editor Report · Decision Letter 1]

5 Aug 2024

Association between nocturia and serum albumin in the U.S. adults from NHANES 2005-2012

PONE-D-24-23292R1

Dear Dr. ,Huimin Long

We’re pleased to inform you that your manuscript has been judged scientifically suitable for publication and will be formally accepted for publication once it meets all outstanding technical requirements.

Kind regards,

Innocent Ijezie Chukwuonye, MBBS, FMCP (Internal Medicine)

Academic Editor

PLOS ONE

---

## [Editor Report · Acceptance letter]

7 Aug 2024

PONE-D-24-23292R1 

PLOS ONE

Dear Dr. Jia, 

I'm pleased to inform you that your manuscript has been deemed suitable for publication in PLOS ONE. Congratulations! Your manuscript is now being handed over to our production team.

Kind regards, 

on behalf of

Dr. Innocent Ijezie Chukwuonye 

Academic Editor

PLOS ONE